# Trust-Aware Routing Mechanism through an Edge Node for IoT-Enabled Sensor Networks

**DOI:** 10.3390/s22207820

**Published:** 2022-10-14

**Authors:** Alaa Saleh, Pallavi Joshi, Rajkumar Singh Rathore, Sandeep Singh Sengar

**Affiliations:** 1General Organization of Remote Sensing (GORS), Latakia P.O. Box 12586, Syria; 2Department of Electronics and Telecommunication Engineering, Faculty of Electronics and Telecommunication Engineering, GH Raisoni Institute of Engineering and Technology, Nagpur 440028, India; 3Department of Computer Science, Cardiff School of Technologies, Cardiff Metropolitan University, Llandaff Campus, Cardiff CF5 2YB, UK

**Keywords:** Internet of Things, modified gray wolf optimization, artificial bee colony optimization, trustworthy nodes, edge node

## Abstract

Although IoT technology is advanced, wireless systems are prone to faults and attacks. The replaying information about routing in the case of multi-hop routing has led to the problem of identity deception among nodes. The devastating attacks against the routing protocols as well as harsh network conditions make the situation even worse. Although most of the research in the literature aim at making the IoT system more trustworthy and ensuring faultlessness, it is still a challenging task. Motivated by this, the present proposal introduces a trust-aware routing mechanism (TARM), which uses an edge node with mobility feature that can collect data from faultless nodes. The edge node works based on a trust evaluation method, which segregates the faulty and anomalous nodes from normal nodes. In TARM, a modified gray wolf optimization (GWO) is used for forming the clusters out of the deployed sensor nodes. Once the clusters are formed, each cluster’s trust values are calculated, and the edge node starts collecting data only from trustworthy nodes via the respective cluster heads. The artificial bee colony optimization algorithm executes the optimal routing path from the trustworthy nodes to the mobile edge node. The simulations show that the proposed method exhibits around a 58% hike in trustworthiness, ensuring the high security offered by the proposed trust evaluation scheme when validated with other similar approaches. It also shows a detection rate of 96.7% in detecting untrustworthy nodes. Additionally, the accuracy of the proposed method reaches 91.96%, which is recorded to be the highest among the similar latest schemes. The performance of the proposed approach has proved that it has overcome many weaknesses of previous similar techniques with low cost and mitigated complexity.

## 1. Introduction

Incredible development in the Internet of Things (IoT) in a decade has led this technology to flourish in various domains. Connecting billions of devices in the physical world through IoT technology has narrowed the gap between the physical and the cyber world. Though many IoT applications have made our lives simpler, it is cumbersome to manage these systems with high security and fidelity. Security and privacy issues are significant concerns to IoT applications [1]. These concerns are so because, due to the unattended randomly placed sensor nodes that are vulnerable to intrusions and attacks, even the data transmitted by sensor nodes are not trustworthy many times. The breach in security may incur huge losses to these systems. Thus, it is indispensable to inhibit such losses to construct a reliable and trustworthy approach [2,3]. Secured data transmission is one of the key challenges in IoT applications. The devices involved in the process of data collection and transmission are more prone to attacks and faults. Some of the applications’ requirements are not only limited to security but also mandate the mitigation of delay, energy usage, and packet loss. Some of the issues such as congestion, delay in transmission, and energy consumption can be dealt with using existing routing algorithms [4,5,6]. Figure 1 shows the cloud-based sensor network having EDGE nodes to send the data to the cloud server for real-time data analysis.

Edge technology, which serves as an extension of the cloud, shows an enormous technological improvement by shifting the burden of storage and computing to the edge network that is confined to an edge layer near the sensing devices. This technique not only alleviates the resource constraints at the end devices but also uplifts the system performance [7]. The Edge network can be easily integrated with IoT networks and achieves a remarkable success in obtaining desired QoS parameters. The trust issues can also be resolved using this technology [8]. There has been enormous research conducted to solve the issue of security by using edge node approaches and trust mechanisms such as [9,10,11]. In [10], the best route path with trustworthy nodes is generated and a mobile edge node visits each sensor with a significant trust value and collects data. Some research works have focused on establishing route paths using trust-based schemes. In [11], a belief-based trust approach to identify malicious nodes is proposed that guarantees defense against Denial of service (DoS), Bad-mouth, and On–Off attacks. The direct and indirect trust values are gathered using the Bayesian estimation technique to determine the correlation in time series data, which can be used to estimate inaccurate knowledge thereby avoiding data from malicious nodes. The issues encountered while designing a network for data collection using trust-based scenarios such as cross-layer optimized networks, 3D underwater sensor networks, UAV-based edge networks, etc. are also major concerns while designing a sensor network. These issues are addressed and resolved in [10,12,13,14].

Although these methods promise good gain and robustness to the system, some loopholes exist in the functionality of these systems. The trust-based routing and selection of trustworthy nodes consumes much energy and time. This paper presents an optimal way of solving this problem by introducing a novel Trust-aware Routing Mechanism. It deals with the IoT architecture exhibiting the edge layer, which uses security-enabled designs, authorization, authentication, privacy, and intrusion detection mechanisms and protocols. It also employs an optimized way to organize the nodes and apply a trust mechanism to all nodes. Furthermore, it establishes a route path using artificial bee colony (ABC) optimization, which is specially designed for edge nodes. The motivation behind this architecture is to provide a buffer facility and enhance the system’s privacy using edge nodes. Our proposed approach is efficient in conserving energy and boosting the accuracy of the system. The main and original contributions of this work are listed below:1.The sensor network having sensor nodes randomly deployed is clustered and the selection of cluster head is conducted using the modified gray wolf optimization approach [15].2.The clustered network consists of many clusters with their respective cluster heads and a mobile edge node that evaluates the trust of every node in a cluster and gathers trustworthy information from those nodes.3.The inter- and intra-cluster routing is optimized using the artificial bee colony optimization approach, which provides the best optimal path for the data flow to the edge node.4.The performance analysis for the proposed model is based on energy consumption, delay, detection rate, accuracy, and network lifetime.

The remainder of this paper is arranged as per the sections mentioned below:

Section 2 gives an overview of the relevant works associated with the technique used in the proposed model. Section 3 describes the problem formulation. A thorough explanation of the techniques used to build the model is discussed in Section 4. It also presents the algorithm of the proposed approach. Section 5 analyses the performance of the work on various metrics and compares it with other similar existing approaches. Section 6 gives a conclusion with a future direction to the proposed methodology.

## 2. Related Works

In this section, we provide an overview of the recently published trust-aware routing protocols with their advantages and limitations. We summarize these in Table 1.

There are many research contributions on trust-aware routing in the literature; we summarize the most recent ones in this section. Wang et al. [16] proposed a trust evaluation scheme based on mobile edge nodes for IoT. In [17], the authors propose a routing protocol called trust-based secure and energy (TBSEER) routing protocol to resolve the congestion and serious energy dissipation problems. In this proposal, the nodes calculate the adaptive direct trust value, and the sink nodes take care of indirect trust computation. Here, the edge nodes efficiently identify the untrustworthy nodes, thereby reducing the risk of attacks. A virtual force data collection (VFDC) scheme is proposed in [10] that enables a mobile edge node to visit every trustworthy node and collect their data to ensure both energy-saving and security of the system. Another approach called Intelligent Trust Cloud (ITCM) [18] management is used to determine the untrustworthy devices in the Internet of Medical Things (IoMT). This approach works in three phases: firstly, the trust clouds are formed; then, they are established through the Fuzzy inference system; finally, the third phase deals with seeking untrustworthy nodes in it.

In [19,20,21], the authors propose trust-based algorithms to select trustworthy nodes, and the data are collected from those nodes using edge node or edge intelligence. These approaches isolate the detected malicious nodes using different techniques. Deep reinforcement learning-based trustworthy target tracking is presented in [8]. This approach is efficient but computationally heavy and resource hungry. The trustworthiness is also helpful in Industrial IoT for secure applications [28]. In addition to the trust evaluation of the sensor nodes, the routing of nodes is another important aspect of the cloud-enabled sensor network. There have been many optimization algorithms implemented by researchers to procure optimal routing for the nodes.

The use of optimization also plays a vital role in establishing optimal paths for the nodes in IoT-based sensor networks. The hybrid algorithms for clustering, namely, K-means with Ant lion optimization (ALO-K) and modified bee colony optimization with K-means, are proposed in [22,23], respectively. In [29], Black Widow (BWO) optimization and ABC protocol are utilized for optimized clustering and routing in a resource-limited sensor network for IoT applications. In [24], the authors use the modified version of artificial bee (ABC) colony optimization called parallel ABC algorithm to detect various attacks and ensure successful communication in edge-enabled industrial IoT. A blockchain approach is used in [26] to address the trustworthiness of IoT agriculture applications in [30]. This approach is cost-efficient but computationally complex. Similarly, ref. [31] also used blockchain technology to address trustworthiness in digital twins. In [25], authors use the Ant Colony Optimization approach (ACO) to fetch data from the rendezvous points in an optimally routed sensor network that generates uneven data from the sensors. In [26], authors use a hybrid approach by implementing Principal Component Analysis (PCA) along with the gray wolf optimizer to optimize the parameters of the Deep neural network, which enables detecting the intrusion in the Internet of Medical Things (IoMT). The authors in [27] incorporate a hybrid metaheuristic approach to choose optimal cluster heads to obtain energy efficiency in the network. This approach uses a hybrid algorithm called Whale Optimization and Simulated Annealing (WOA-SA).

The above-discussed methods deal with the problem of security and energy consumption by providing various solutions such as trust-based data collection, optimized routing, and edge-node-enabled secure data transmission. However, a trade-off has been observed between the efficiency and security of the algorithms, as mentioned above. This is because some researchers use the edge nodes for secure and prosperous transmission based on the trust values of nodes, and some use optimized routing to enhance the system’s QoS parameters. So, a hybrid approach that uses edge node computing and optimized route establishment for data transmission needs to be developed in clustered IoT-enabled sensor networks. This paper uses a modified gray wolf optimizer (GWO) for the clustering process. It provides high convergence with no local optima problem. It also employs the trust evaluation approach to pick trustworthy nodes. The route path is devised by implementing the artificial bee colony optimizer, enabling the mobile edge node to collect data optimally from all trustworthy nodes.

## 3. Problem Statement

The goal of this article is to evaluate the trustworthy nodes and to detect the faulty nodes using the proposed trust evaluation scheme on the nodes. This improves the security of the IoT cloud. We assume the highest energy mobile node as an EDGE node called ‘MEN’, which is capable of collecting and fusing the data to the sensor cloud network. It also provides good computing power and a buffer facility for unused data. The set of sensors deployed in a $50\times 50\;m^{2}$ underlying cloud-based sensor network are $sn=\{sn_{1},sn_{2},\ldots ,sn_{n}\}$. The trusted nodes are identified by calculating the trust values considering various parameters. $I^{dir}$ and $I^{indir}$ are direct and indirect trust values used to evaluate the trustworthiness of the nodes in each cluster. Furthermore, the trusted nodes from every cluster are considered to route via trust paths for optimal data aggregation at the MEN.

## 4. Proposed Trust-Aware Routing Mechanism

This paper deals with trust issues arising during data transmission in IoT-enabled WSNs. The proposed model uses an edge node to fetch the data from only trusted nodes. However, detecting the trusted nodes is a challenging task. In this context, we proposed a Trust-Aware Routing Mechanism (TARM) through an edge node for IoT-enabled WSNs, which is summarized using a flow graph in Figure 2. The proposed algorithm uses modified GWO and ABC optimizations to achieve the goal in this context. Modified GWO is employed for the clustering of nodes to ensure energy efficiency in the network. Then, all nodes of each cluster are checked for their trustworthiness using a trust evaluation algorithm based on the calculation of direct and indirect trust values. Once the untrustworthy nodes are identified, the mobile edge node collects the data from all CHs in each cluster. The notations used in the paper are given along with their meanings in Table 2.

An optimized routing mechanism based on ABC optimization is utilized to efficiently route the data from nodes to CH and from CH to Mobile Edge Node (MEN). The ABC aims to find an optimized relay path from the nodes that reach CH and ultimately the MEN. There are three sets of bees—onlookers, scouts, and employee bees. The suggested ABC approach for routing guarantees the drop in energy depletion during the routing of information in inter- and intra-cluster. The low hop count indicates that this is a low energy consumption when data are sent from nodes to CH and all CHs to the mobile edge node. The process is repeated, and scout bees search and forage the food site again. Figure 3 depicts the routing paths for the nodes, which are evaluated as trustworthy and untrustworthy nodes, the latter of which do not participate in data transmission.

### 4.1. Clustering through Modified GWO

The modified GWO [15] is employed to elect the cluster head (CH) in every cluster. The distance, CH balancing factor, and residual energy are the primary factors to identify the best CH in a cluster. The positions of the nodes are sent to the BS after the network deployment. The BS implements a modified GWO algorithm (Algorithm 1) using the objective function based on the above parameters. Initially, for the sensor nodes whose residual energy is higher by 10% compared with average energy, those nodes are considered to be CHs. Based on the hierarchy of dominance of gray wolves (high-energy nodes), α is considered the best solution, (with the least value of fitness) followed by β (higher value than the least) and then δ. Apart from those, ω is taken as the least optimal solution calculated by the objective function given below:(1)$$\begin{array}{c} F=\left(t-z\right)\times O_{1}+\left(1-x+t\right)\times O_{2}\\ +\left(1-t\right)O_{3}+\left(t-y\right)\times O_{4} \end{array}$$
where t=y+z,x+t=1
(2)$$O_{1}=\frac{1}{\sum_{a=1}^{b}\left(E_{{CH}_{a}}\right)}$$

The above equation represents the total energy of all CHs. A minimum value of this factor denotes less energy usage, where *a* is the cluster number and *b* denotes total CHs. ${CH}_{a}$ represents cluster head in cluster *a*.
(3)$$O_{2}=\sum_{a=1}^{b}\frac{n}{b}-c_{a}$$
$O_{2}$ is the balancing factor. To avoid the formation of some huge and small clusters, this parameter is defined to balance the cluster size to minimize power consumption. *n* is the total number of sensor nodes and $c_{a}$ represents the number of sensor nodes in cluster *a*.
(4)$$O_{3}=\sum_{a=1}^{b}\frac{1}{c_{a}}-dist\left({CH}_{a},\;S\right)$$
$O_{3}$ is the average sink distance from each CH. *S* is the base station position.
(5)$$O_{4}=\sum_{a=1}^{b}\frac{1}{c_{a}}\sum_{k=1}^{c_{a}}dist\left({sn}_{k},{CH}_{a}\right)$$
$O_{4}$ is the average intra-cluster distance and dist(${sn}_{k},\;{CH}_{a}$) is the total intra-cluster distance of each node from its CH. ${sn}_{k}$ are ‘*k*’ sensors that are in a communication range. The above equation must evaluate the minimum value. The overall fitness function defined in Equation (Equation 1) needs to be minimized. For an optimal selection of CH, the above objective function is evaluated for each CH. The node with the least value of fitness in that cluster is selected as CH.

While hunting, gray wolves surround their prey and then attack it. Initially, the location of the prey is unknown. The hunting process is governed by best solutions alpha, beta, and omega and they upgrade their positions using the search agents beta and alpha, shown in Equations (Equation 6)–(Equation 12).
(6)$${\vec{A}}_{\alpha }=\left|{\vec{B}}_{1}\cdot {\vec{P}}_{\alpha }-\vec{P}\right|$$
(7)$${\vec{A}}_{\beta }=\left|{\vec{B}}_{2}\cdot {\vec{P}}_{\beta }-\vec{P}\right|$$
(8)$${\vec{A}}_{\delta }=\left|{\vec{B}}_{3}\cdot {\vec{P}}_{\delta }-\vec{P}\right|$$
(9)$${\vec{P}}_{1}={\vec{P}}_{\alpha }-{\vec{C}}_{1}\cdot {(\vec{A}}_{\alpha })$$
(10)$${\vec{P}}_{2}={\vec{P}}_{\beta }-{\vec{C}}_{2}\cdot \left({\vec{A}}_{\beta }\right)$$
(11)$${\vec{P}}_{3}={\vec{P}}_{\delta }-{\vec{C}}_{3}\cdot \left({\vec{A}}_{\delta }\right)$$
(12)$$P\left(p+1\right)=\frac{{\vec{P}}_{1}+{\vec{P}}_{2}+{\vec{P}}_{3}}{3}$$
where ${\vec{P}}_{n}$ is the position of CH; n=1,2…,${\vec{P}}_{\alpha }$, ${\vec{P}}_{\beta }$, ${\vec{P}}_{\delta }$ are the positions of gray wolves (best optimal solutions); *p* is the ongoing iteration; ${\vec{B}}_{n}$, ${\vec{C}}_{n}$ are coefficient vectors for various prey positions, where $\vec{B}=2\cdot \vec{m_{1}}$ and $\vec{C}=2\vec{a}\cdot \vec{m_{2}}-\vec{a}$. Here, $\vec{m_{1}}$ and $\vec{m_{2}}$ are vectors randomly picked between [0, 1].

**Algorithm 1.** Modified gray wolf optimizer based clusteringInput: pop, pmax, $P_{\alpha }$, $P_{\beta }$, $P_{\delta }$
Output: $P_{\alpha }$  1: Initialize the gray wolf population $P_{n}$ randomly  2: Initialize a, B & C  3: Determine the fitness of $P_{n}$ using Equation (Equation 1)  4: **while**
i≤pmax
**do**  5:    **for** each wolf $P_{n}$ **do**  6:       Update the position using Equation (Equation 12)  7:       Update $P_{\alpha }$, $P_{\beta }$, $P_{\delta }$  8:       i=i+1  9:    **end for**  10: **end while**  11: **return**
$P_{\alpha }$

Finally, the wolves attack, and the range of $\vec{C}$ also reduces in [−2a,2a], where *a* is reduced to 0 from 2. The two conditions that play an important role in selecting the optimal cluster head are as follows:(1)When $\left|C\right|>1$, the nodes diverge to search for the prey depending on the positions of α, β, and δ.(2)When $\left|C\right|<1$, the wolves converge to attack the prey. After the successful completion of clustering, the optimal CHs are chosen and the non-CHs join the CH in their proximity by sending a request for joining as a cluster member.

After successful completion of clustering, the optimal CHs are chosen and the non-CHs join the CH in their proximity by sending a request for joining as a cluster member. The non-CHs are acknowledged by receiving an acceptance; this way, the clusters are formed. Figure 4 shows the positions of α, β, δ wolves and the candidate wolf with reference to the position of the prey. The nearest position is of α.

### 4.2. Trust-Value-Based Route Path Selection

In this section, we evaluate the values for trusted nodes along with the routing strategy.

#### 4.2.1. Trust Values Evaluation

To ensure data trustworthiness, nodes are at first checked whether they are trustworthy or not. The mobile edge node, which gathers information, evaluates the node trust, thereby reducing the distance and mobile delays. Let $I_{t}$ be the initial trustworthiness of nodes, *T* be the threshold for trustworthiness, and $I_{c}$ be the trust value of a node. Keeping the storage space of all nodes in consideration, the space for memory can be taken in [0, 10] intervals. Let 5 be the initial trust value of the node. If $I_{c}=0$, the node is untrustworthy. If $I_{c}=10$, it shows complete trustworthiness. To avoid the biased performance of nodes, a multi-dimensional model for evaluating trust is employed. The trust calculation exhibits direct and indirect trusts, which is discussed as follows:

**DIRECT TRUST**: Manifests a trust relationship between directly communicating nodes. The direct trust values can be determined using the below parameters [21]:
1.Communication trust value:
(13)$$I_{com}^{dir}=v_{old}\times I_{old}^{dir}+v_{new}\times I_{new}^{dir}$$
(14)$$I_{new}^{dir}=\frac{SC}{N}$$$I_{com}^{dir}$ is the communication trust value; $v_{old}$ and $v_{new}$ are the weight variables of old and new trust values, respectively; $I_{old}^{dir}$ and $I_{new}^{dir}$ are old and new trust values, respectively. SC is the number of successful communications and *N* is the total communications.2.Positional intimacy: The measure of successful delivery of messages when a node is very close to CH.
(15)$$I_{c}^{dir}=1-\frac{dist(c,\;CH)}{R}$$$I_{c}^{dir}$ indicates trust value of the node with location, dist $\left(c,\;CH\right)$ is the distance between the node and CH, *R* is the range of communication for that node.3.Loss of packets:
(16)$$I_{p}^{dir}=\frac{S_{p}-R_{p}}{S_{p}}$$$S_{p}$ is the number of packets sent and $R_{p}$ expresses the number of packets received.4.Energy: Determines the lifetime of nodes. Trust value of a node in terms of energy is the ratio of its residual energy to its initial energy
(17)$$I_{i,j}^{dir}=v_{com}\times I_{c}+v_{l}\times I_{l}+v_{e}\times I_{e}+v_{p}\times I_{p}$$$I_{i,j}^{dir}$ shows the total trust values of *i* and *j*; $v_{com}$, $v_{l}$, $v_{e}$, and $v_{p}$ are the weights of communication trust; and $v_{com}+v_{l}+v_{e}+v_{p}=1$.

INDIRECT TRUST: When two nodes communicate via a recommendation of a third node adjacent to one of the two nodes, the indirect trust value has to be calculated. For example, *A* and *B* nodes interact via intermediary nodes *x*, *y*, and *z*, which are the neighborhood nodes to *A*. Then, A projects indirect trust at *B*, and the trust is transferred from *A* to *B* through nodes *x*, *y*, and *z*. The indirect trust of node *A* at *B* can be expressed as
(18)$${{(I}_{A,B})}_{x}=I_{A,\;x}\times I_{x,B}$$
${{(I}_{A,B})}_{x}$ is the trust of A after passing through x to B. The indirect trust of *A* at *B* is expressed as
(19)$$I_{i,j}^{indir}=I_{A,\;B}=\sum_{x=1}^{m}\partial_{x}\times {\left(I_{A,\;B}\right)}_{x}$$
(20)$$\partial_{x}=\left[\frac{\left(I_{x,B}\right)}{\sum_{n=1}^{m}I_{x,n}}\right]$$
where *n* is the counter representing *m* adjacent nodes to *A*. The total trust is given by the sum of direct trust ($I^{dir}$) and indirect trust ($I^{indir}$), which is calculated as
(21)$$Tot\_trust=I^{dir}+I^{indir}$$

To identify the faulty node and trustworthy node, the following measure should be considered:(22)$$Trust=\left\{\begin{array}{cc} trusted\_node & \mathrm{if}\;Tot\_trust=1\\ faulty\_node & \mathrm{if}\;Tot\_trust=0 \end{array}\right.$$

Algorithm 2 shows how nodes are evaluated based on trust values and how untrustworthy nodes are identified. The trusted routing path based on the optimized ABC approach, utilized to transfer data from all trusted nodes to all CHs and then to the MEN, can be achieved with Algorithm 3. Figure 2 represents the flow of the proposed methodology.
**Algorithm 2.** Trust evaluation schemeInput: W, $CH_{i}$, MEN, ROutput: Trustworthy nodes with trust values for each cluster, optimized inter- and intra-cluster route paths  1: **for** all nodes in *c* cluster **do**  2:    **if** dist(currentnodeA,CH)<R **then**  3:      the node is directly communicated and compute direct trust value using Equations (Equation 13)–(Equation 17)  4:    **else**  5:      The node is not directly communicated  6:      **for** *A* node **do**  7:         **if** dist(Atox+xtoCH)<R **then**  8:           Mark *x* as the neighbor of *A* and compute indirect trust values using Equation (Equation 19)  9:         **else**  10:           node *x* is not the neighbor of *A*  11:         **end if**  12:      **end for**  13:    **end if**  14: **end for**

**Algorithm 3.** ABC routing on trusted paths for all trusted neighboring nodes
1:The scout bee (cluster member) searches for food resources2:Initiating food source $\left(F_{ij}\right)$3:Finding optimized solutions (relay CH) for food sites by employee bees using Equation (Equation 19)4:The CH transmits “PATH_MESSAGE” (which includes energy, node ID, length of queue, etc.) to the nodes in its cluster.5:New food source by employee bee is generated using Equation (Equation 26)6:Employee bee shares all best solutions with onlooker bees7:The onlooker bee chooses the best solution using the probability $p_{i}$ elucidated in Equation (Equation 28)8:**if** MEN receive the data packets **then**9:   It acknowledges the CH intending the delivery probability. The upgrade in delivery can be achieved using Equation (Equation 29)10:
**end if**
11:**for** All paths in the network **do**12:   **if** the present source exceeds the limit of the search **then**13:     it is not considered as an optimized solution and is discarded.14:   **else**15:     CHs transmit their data to the MEN using the optimized path16:   **end if**17:
**end for**



#### 4.2.2. Routing

The artificial bee colony algorithm [32] is a search-based probabilistic evolutionary approach of optimization. It works in three stages: (i) food sources; (ii) foraging for food by employer bees; (iii) observing and random searching by scout and onlooker bees to know the food source, which is the solution set. Although the search time of this algorithm is longer, it is beneficial in preventing premature convergence. To combat the problem of search time, we have modified the fitness function and developed the ABC approach, which is inspired by [24] to enhance its accuracy. The following steps can be followed to easily understand the proposed ABC approach:1.Initialize the swarm size, food source, employer bees, and onlooker bees such that $S=Emp=Onl=\frac{W}{2}$. The scout bee (cluster member) searches for food resources ${CH}_{i}$, $r=f_{i},\;1,f_{i},\;2,\ldots f_{i},R$2.Initialize the random solutions using the equation below:
(23)$$F_{ij}=F_{j}^{min}+rand\left(0,1\right)\times \left(F_{j}^{max}-F_{j}^{min}\right)$$3.Find optimized solutions (relay CH) for food sites by employee bees using the equation below:
(24)$$\begin{array}{c} {RG}_{CH}\left({ch}_{j}\right)={ch}_{i}|dist\left({ch}_{j}{,ch}_{i}\right)\leq \min RG\left({ch}_{j}\right)\\ ;dist\left({ch}_{i},MEN\right)<dist\left({ch}_{j},MEN\right) \end{array}$$The expression for objective function is given as
(25)$$f_{i}\left(\right)=\left(1-t\right)O_{3}+\left(t-y\right)\times \;O_{4}\;$$
where $O_{3}$ and $O_{4}$ are the same as shown in Equation (Equation 4) and Equation (Equation 5), respectively. The fitness for every solution is calculated using
(26)$$Fit=\left\{\begin{array}{cc} \frac{1}{1+f_{i}\left(\right)} & \mathrm{if}\;f_{i}\left(\right)\geq 0\\ 1+|f_{i}\left(\right)| & \mathrm{if}\;f_{i}\left(\right)<0 \end{array}\right.$$4.The random solutions are generated from the above equation, this solution set chooses candidate and partner solutions from the remainder set. The new solution by selecting a decision variable from the first chosen solution can be generated using Equation (Equation 27)
(27)$$\begin{array}{c} F_{ij}^{new}=F_{ij}\times rand\left(0,1\right)+\left(F_{ij}-F_{kj}\right)\varphi_{ij}+\\ \left(z_{ij}-F_{ij}\right){\left(\theta_{ij}-0.5\right)}^{2} \end{array}$$
where $\left(z_{ij}-F_{ij}\right){\left(\theta_{ij}-0.5\right)}^{2}$ is the global best solution that produces optimal convergence efficiently and $\theta_{ij}$ is the new step size given by $\theta_{ij}=1-\frac{itr_{c}}{t}$. The above Equation (Equation 26) enables the CHs to quickly relay the data to the nearest trustworthy CH or node.5.The next phase is the onlooker bee phase; these onlooker bees are managed by the CH nodes. It searches for the relay CH depending on the following probability for each solution:
(28)$$p_{i}=\frac{o_{i}}{\sum_{ch\in {RG}_{{CH}_{\left({ch}_{j}\right)}}}o_{i}}$$
where $o_{i}=\frac{w_{j}}{\sum ch\in {RG}_{{CH}_{\left({ch}_{j}\right)}}w_{s}}\times \frac{{DD}_{i}}{\sum_{h\in {RG}_{{CH}_{\left({ch}_{j}\right)}}}{DD}_{s}}$. Once all the probabilities associated with every solution are obtained, the onlooker bees select the initial *S* and compare its probability with the random number x∈[0,1].(a)If $x>p_{i}$, the onlooker bees repeat the same process and do not generate new *S*.(b)If $x<p_{i}$, the onlookers generate new *S* by choosing an arbitrary decision variable in current *S* using Equation (Equation 26).6.The scout bees use a trail vector to keep track of the failures when they fail to generate a better solution. Here, the new solutions are generated using Equation (Equation 23). The best fitness value among the previous and newly generated solutions based on the fitness function is stored in the trail vector.7.If the mobile edge node receives the data from CH, it acknowledges CH intimating delivery. The delivery update equation is determined by Equation (Equation 29):
(29)$$DD{\left({CH}_{m}\right)}^{\prime }=\frac{DD\left({CH}_{l}\right)+DD\left({CH}_{m}\right)}{2}$$
where *l* and *m* are the two nodes in a cluster.8.Based on the optimized path, the nearest CH transmits its data to the MEN.

The pseudo-code for the trust evaluation scheme based on the ABC routing approach for mobile edge node is provided in Algorithm 1. Figure 5 portrays a visualization of the proposed TARM that focuses on the clustered network wherein clustering is performed by the modified GWO. This clustered network then undergoes a trust evaluation mechanism at MEN to identify trustworthy nodes. Routing through the trustworthy nodes via the CH is then performed using the ABC algorithm explained in Section 4.2.2. It initially evaluates the trust values of the nodes and classifies them as trustworthy and untrustworthy nodes. Then, it applies the ABC approach proposed in this section on the MEN and enables the optimal route discovery for MEN through the trusted nodes and, finally, to the base station. The edge node is used because of its properties such as storage capability, processing ability, and location near the base station. The mobility of the MEN is restricted to its transmission radius. Nevertheless, the CHs are able to relay the information to other CH nodes in order to make the information reach the CH nearer to the MEN.

### 4.3. Complexity Analysis

In general, the GWO algorithm starts with the initialization of parameters with a complexity of $O(n_{w})$ where $n_{w}$ is the number of packs. An outer for loop runs until maximum rounds are reached computing the complexity of $O(n_{w})$. Now, the inner three for loops run until the number of packs are reached and they count the complexity of $O({maxit,n}_{w},s)$, where maxit is the maximum iterations and *s* is the size of the pack. Thus, the total complexity for the worst case is determined by $O\left(n_{w}\right)+O\left(n_{w}\right)+O\left({maxit,n}_{w},s\right)$. ABC does not depend on the problem dimension; thus, time complexity for it is scaled to O(n) where *n* is the size of the bee population. The search complexity (depending on onlooker bee population) of ABC is given by 2×LB×maxcy, where LB is the number of onlooker bees and maxcy indicates maximum cycles.

## 5. Performance Evaluation

In this section, the performance of the proposed TARM is verified in the MATLAB software (8 GB RAM, i7 core CPU Windows 10 system) and is validated with other similar latest trust-based models that employ edge nodes such as ITCM [18], TBSEER [17], VFDC [10], TEC-SFS [33], and TEM-MEN [16]. Deployment of the sensor nodes in the network is considered according to Sah et al. [34]. The simulation is performed using some parameter settings described in Table 3. We evaluated and plotted (Figure 6) different metrics including the rate of detecting the untrustworthy nodes to the total number of nodes in the network, the accuracy of the model with the increasing percentage of untrustworthy nodes, overall energy utilization in the network, the trust values of nodes with increasing untrustworthy node percentage, average end-to-end delay, and network life evaluation.

The experiment includes one IoT network that has clustered WSN scenarios, one MEN, and one base station. The CH in each cluster gathers data from the nodes once the trustworthy nodes are identified. The CH has the ability to relay data to other CHs until it reaches the CH closest to the MEN. The trusted route path established using ABC optimization is updated at the MEN, and the MEN with the data transmission rate of 2 Mbps receives the data from its nearest CH. The data transmission time from MEN to the cloud server is 10 ms. The MEN moves with uniform speed and its motion is restricted to the CH, which is in its proximity. This is performed to save energy and to avoid unnecessary delays. However, the selection of starting point, radius of communication, and speed can be adjusted.

### 5.1. Accuracy and Rate of Detection of Untrustworthy Nodes

Figure 6a presents the comparison of various approaches in terms of detection rate in percentage while varying the number of nodes. When the network undergoes a high number of detection rounds, the detection rate of the proposed TARM technique to detect the untrustworthy nodes enhances gradually. In Figure 6a, the approaches considered for comparison are VFDC, TEM-MEN, TEC-SFS, and the proposed method. We can see that the rate at which the proposed TARM detects the untrustworthy nodes increases rapidly compared with the TEM-MEN and TEC-SFS, which show a more or less constant increase in the rate of detection. However, TEC-SFS has a good detection rate but the rate of increase of the detection percent in the case of TEC-SFS from zero number of nodes to 200 is not much and it gradually increases at a constant pace. Both TEM-MEN and TEC-SFS show only a 14.6% and 9.5% increase in the rate of detection when starting from the 0th node to 200th node, respectively. At the same time, the TARM offers a 67.6% hike in the enhancement of detection rate from 0 nodes to 200 nodes. TARM has a slightly higher detection rate than the VFDC and TEM-MEN for the maximum number of nodes. Further, it is examined that as the iterations of the moving path enhances, the rate of recognition of trust evaluation correspondingly increases.

The accuracy of the trust evaluation mechanism is defined as how accurately the scheme can recognize untrustworthy nodes. It is verified that the TBSEER and proposed method both resist sinkhole attacks under a varying number of malicious nodes. As shown in Figure 6b, the accuracy plot is drawn against the percentage of untrustworthy nodes. The lowest accuracy is shown by the VFDC method, which is 71.1%. The highest accuracy is shown by the proposed method, which is 91.96%. The TBSEER performs very well in this case and records an accuracy of 88.76% for the highest percentage of untrustworthy nodes.

Figure 6c shows the initial route path in the absence of a trust evaluation scheme and the trusted route when the trust evaluation algorithm is executed. Figure 6c shows that the trusted route path takes mostly trustworthy nodes and covers a very short distance to transmit the data to the destination, whereas the same task if performed by the initial route path takes more nodes and hops, which enhances the distance as well as the energy and latency in the network. The calculation for energy consumption for the trusted route has been performed and is represented in Figure 6d.

### 5.2. Impact of the Percentage of Untrustworthy Nodes on QoS Parameters

Figure 6d shows energy consumption in the whole network for the increasing number of untrustworthy nodes. The proposed method, as observed in Figure 6d, outperforms the other three similar methods. The TBSEER method shows 13.09% more energy consumption compared with the proposed method. Similarly, for VFDC, the percentage enhancement in energy usage is 54.92% higher than the energy usage in the proposed method. The proposed method shows 73.37% higher energy efficiency compared with the ITCM approach.

Figure 6e depicts the node trust values for increasing percentage of untrustworthy nodes. As the proportion of the untrustworthy nodes enhances to 50%, the trust values of the nodes decline rapidly. This factor reflects the trustworthiness of the proposed trust evaluation scheme. All the algorithms show the same declination but the proposed method shows superiority over others. The proposed model shows 58% higher trustworthiness compared with the TBSEER method and this percentage goes beyond 100 for other approaches.

In Figure 6f, the value of average end-to-end delay rises as the percentage of untrustworthy nodes increases. In the case of TBSEER, the frequent packet losses are responsible for the delay in establishing the link between the nodes for packet re-transmission. Moreover, it leads to the degradation of the routing stability. The proposed approach shows 85.16% less delay than the TBSEER protocol. The TEM-MEN approach is tested on the cloud as well as edge computing and it is verified that edge computing is more reliable for real-time applications as it offers low latency, as shown in Figure 6f. The ITCM offers 81.77% high delay and the proposed method is 35.5% more delay-efficient than the TEM-MEN approach.

The network lifetime analysis is depicted in Figure 6g. As can be observed from the graph, the network lifetime represented by the number of rounds is calculated when the first node dies, when the half node dies, and when the last node dies. The proposed model offers the highest network life for all three conditions ensuring the effectiveness and reliability of the system over a longer period. The ITCM performs poorly; TBSEER performs moderately in all three cases and outperforms the other two algorithms, which are ITCM and VFDC. The proposed method is 83.8%, 45.1%, and 28% more efficient compared with TBSEER for FND, HND, and LND, respectively. In general, it is verified that for the same network size, the scenario with a high number of IoT devices offers short network life compared with the small scenario. In the proposed method, the network size of $50\times 50\;m^{2}$ is kept constant and the number of nodes is increased. It is recorded that there is not much difference between the scenario with 50 nodes and the scenario with 200 nodes. However, in the case of ITCM, the network life is short because of the same reason.

Figure 6h depicts the packet loss rate for TBSEER, VFDC, TEM-MEN, and the proposed method. It can be seen from the graph that as the percentage of untrustworthy nodes increases in the network, a large number of packets or data flows to the untrustworthy nodes produces a gradual increase in the packet loss rate. However, the proposed trust model offers low packet loss since the distance of the average route length gets reduced resulting in the low untrustworthy nodes encountering the trusted route. It is observed that for a low percentage of untrustworthy nodes, the percentage hike in the packet loss rate is 38.74%, 45.5%, and 59.39% for TEM-MEN, TBSEER, and VFDC methods, respectively, when contrasted with the proposed method. For a higher percentage of trustworthy nodes, the percentage packet loss of the proposed method reduces by 24.86% when compared with TEM-MEN. TBSEER performs better than VFDC and shows an 11.66% decrease in packet loss rate than that in VFDC.

Table 4 gives the average route length for the trust evaluation approaches that employ edge nodes. The proposed scheme calculates the shortest route length for the trusted route. The longest route path is given by the VFDC algorithm. TBSEER performs moderately and the TEM-MEN approach performs well after the proposed method. Table 5 shows the comparison between various optimization approaches on the basis of energy usage, data sent to the cluster heads, and cluster load balance factor. It can be clearly observed that the proposed model, which employs a modified GWO algorithm, delivers the highest number of packets to CH—i.e., 1858—whereas the other optimization approaches such as PSO, Cuckoo search, GWO, and whale optimization, respectively, delivered 1124, 1466, 1689, and 1256 packets successfully to CH node. The minimum load balance factor for cluster balance is recorded for modified GWO and is 0.48. The energy consumed during the intra-cluster communication is lower for modified GWO (0.26 J) than for PSO (0.89 J), GWO (0.38 J), CS (0.73 J), and WOA (0.45 J).

Besides this, a separate set of experiments are conducted on the two untrusted scenarios, which are mentioned below:1.Of the total 200 nodes, 25% or 50 untrustworthy nodes are considered to be uniformly distributed in the given area of $50\times 50\;m^{2}$.2.Of the total 200 nodes, 50% or 100 untrustworthy nodes are considered to be uniformly distributed in the given area of $50\times 50\;m^{2}$.

The simulation results for network lifetime, throughput analysis, and energy efficiency are presented in Figure 7, Figure 8 and Figure 9, respectively. The proposed method is compared against the untrusted scenarios of TBSEER and VFDC approaches. Figure 7a depicts the network life for the 25% of untrustworthy nodes out of the total 200 nodes. Similarly, Figure 7b gives the network lifetime analysis for 50% of untrustworthy nodes out of the total 200 nodes. It is noticed that the network lifetime of the proposed TARM method is higher than the other two methods. Figure 8 presents the throughput evaluation for 25% and 50% of untrustworthy nodes out of 200 nodes in Figure 8a,b, respectively.

Finally, the efficiency in terms of energy is depicted for both the untrusted scenarios against TBSEER and VFDC methods in Figure 9a,b, respectively. It is observed that for the first scenario—that is, 25% of untrustworthy nodes out of total nodes, the performance of the proposed method is better and sustains for more iterations compared with the second scenario—that is, 50% of untrustworthy nodes out of 200 nodes. This shows that as the number of untrustworthy nodes increases, the performance starts degrading. Hence, the proposed method works on detecting the faulty or untrustworthy nodes and establishing trust routes using trusted nodes in the sensor network.

### 5.3. Statistical Evaluation

The statistical evaluation is the non-parametric evaluation of the approach performed for validating the performance of the proposed technique with other similar techniques. For comparison, we consider five existing but similar approaches i.e., particle swarm optimization (PSO), ant colony optimization (ACO), hybrid whale GWO [35], modified bee colony optimization (MBCO) [23], and ant lion optimization with K-means (ALO-K) [22]. The objective function is taken to be the proposed one, which is expressed in Equation (Equation 1). Table 6 shows the success rates and average time taken for execution of all these algorithms. Higher success rate exhibits better performance, and lower time taken for execution leads to faster convergence of the algorithm. It can be noted that the proposed modified GWO and ABC algorithms performs better than other existing algorithms.

We compare the heuristic algorithms using the datasets described in Table 7 and paired observations. Statistical tests such as *Friedman’s* test and *Holm’s* test help in identifying the best heuristic approach for trustworthiness. The test procedures used in this paper to analyze the observations in datasets are Friedman’s test and Holm’s test. In these methods, the null hypothesis $\left(H_{0}\right)$ says that all algorithms have the same variances and mean cost while the alternate hypothesis $\left(H_{1}\right)$ says that all algorithms have different variances and mean cost.

#### 5.3.1. Friedman Test

It is used to find the variations between the groups which have rank-dependent variables. The null hypothesis states that all algorithms perform equally while the alternate hypothesis says that there are differences in each algorithm’s performance. Here, we consider the best optimal values of intra-cluster distances expressed in Equation (Equation 5) as the basis to assign ranks (written in brackets after each best-optimized value) from one to six, as shown in Table 8. The Friedman distribution value is given by
(30)$$Fr=\frac{\left(x-1\right)\chi_{Fr}^{2}}{x\left(y-1\right)-\chi_{Fr}^{2}}$$
where $\chi_{Fr}^{2}$ is expressed as
(31)$$\chi_{Fr}^{2}=\frac{12x}{y(y+1)}\left[\sum_{i=1}^{y}r_{i}^{2}-\frac{y{\left(y+1\right)}^{2}}{4}\right]$$
where *x* is the number of datasets and *y* is the total number of algorithms (x=5 and y=6) being compared. The degree of freedom is (y−1) and (y−1)(x−1), i.e., between 5 and 20; using this, the critical value obtained from *F*-distribution at α=0.05 is 2.711. The values of $\chi_{Fr}^{2}$ and Fr computed from the above equations are 18.428 and 11.216, respectively. As it can be noticed that Fr is greater than the critical value, the null hypothesis is rejected, showing that the performances of all algorithms are different.

#### 5.3.2. Holm’s Test

Friedman’s test shows the rejection of the null hypothesis for all algorithms; so, here, we use Holm’s procedure as a post-hoc test. The *z*-value is given by $z=\frac{r_{0}-r_{i}}{h}$, where the ranks are taken from Table 8. $r_{0}$ is the average rank of the proposed method and $h=\sqrt{\frac{y(y+1)}{6x}}$. Using this *z*-value, the *p*-value is obtained from a normal distribution chart. This *p*-value is compared with $\frac{\alpha }{\left(y-i\right)}$. If the *p*-value is less than $\frac{\alpha }{\left(y-i\right)}$, the null hypothesis is rejected; otherwise, it is accepted. Table 9 shows that Hybrid WGWO, MBCO, and ALO-K perform similarly to the proposed algorithm and other heuristic approaches perform differently.

## 6. Conclusions

To establish a trust-based relationship between the nodes in an IoT-based network and a mobile edge data collector for dynamic scenarios, we develop a trust-aware model that enables clustering and routing in a sensor network. In our proposed work, an edge-enabled IoT architecture is implemented to find trustworthy nodes and those nodes are only targeted for data gathering. The clustering is performed using modified gray wolf optimization and the ABC approach is utilized for effective routing. The results manifested that the proposed work ensures energy-saving and has the advantage of having negligible packets drop. The proposed scheme shows approximately 80% reduction in average delay and around 50–70% reduction in energy consumption when compared with other state-of-the-art approaches. Furthermore, the accuracy of the proposed scheme is recorded to be 91.96%. The proposed distributed method can be easily implemented for a broad range of IoT applications that ensures good data transmission and information storage by virtue of edge intelligence. As a future direction, this work can be further extended for large-scale sensor networks that can use more than one edge node. The method can be made computationally efficient by employing some novel learning techniques to find trustworthy nodes and routes. Furthermore, multi-objective optimization can also be used to achieve energy-efficient optimized clustering and routing.

## Figures and Tables

**Figure 1 sensors-22-07820-f001:**
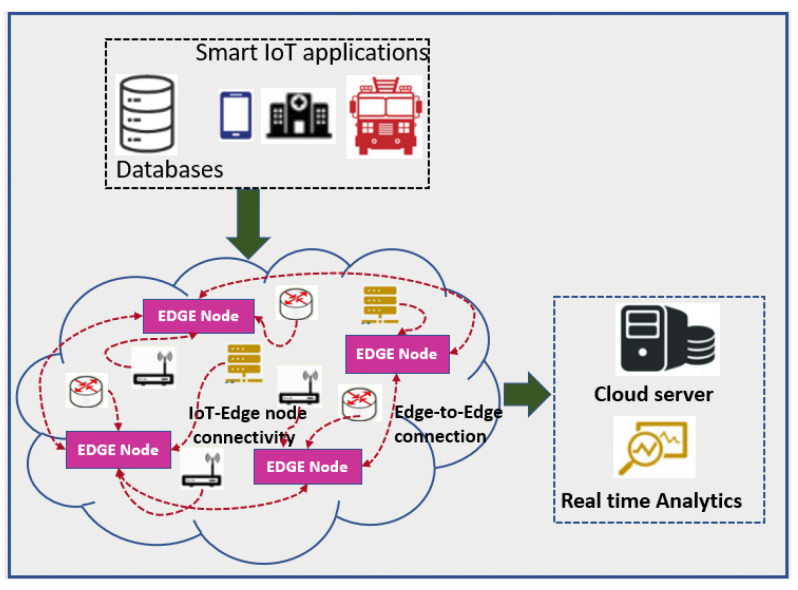
EDGE-node-based IoT application.

**Figure 2 sensors-22-07820-f002:**
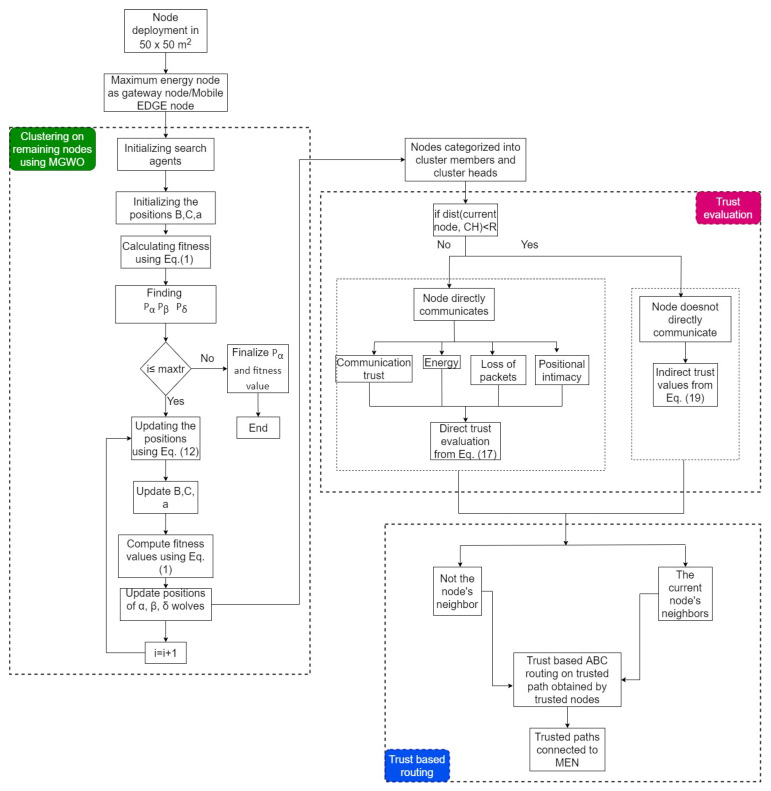
Optimized clustering and trust evaluation process with ABC-based routing.

**Figure 3 sensors-22-07820-f003:**
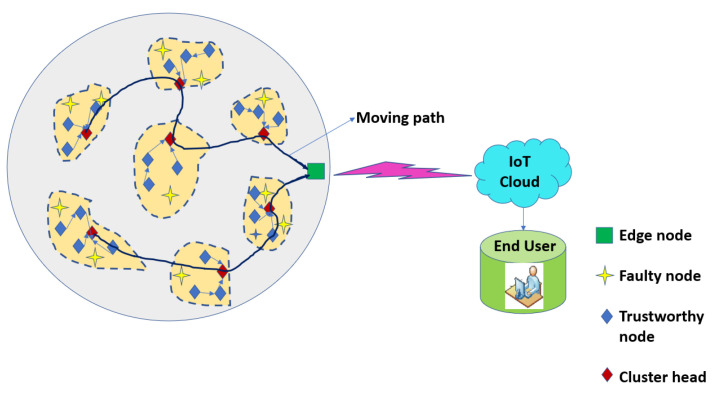
Evaluation of trustworthy nodes establishing optimized trust path for data routing.

**Figure 4 sensors-22-07820-f004:**
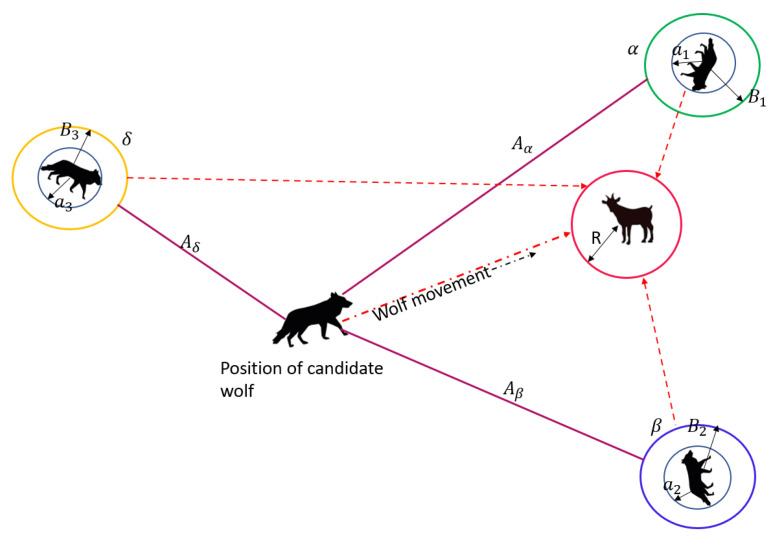
α, β, δ, and candidate wolves with reference to the position of the prey.

**Figure 5 sensors-22-07820-f005:**
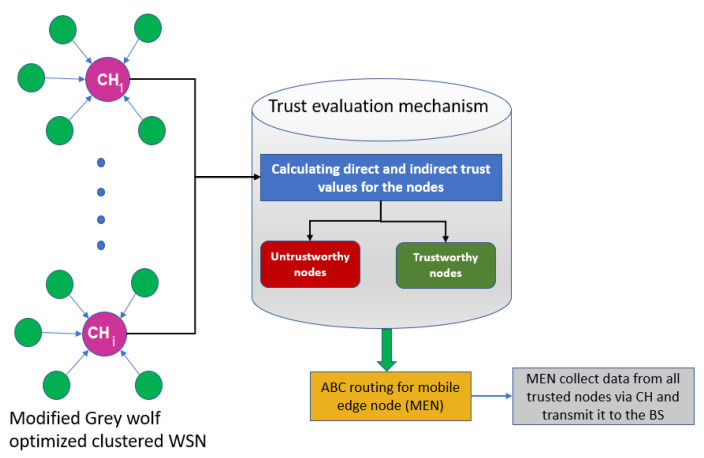
The trust evaluation model for proposed TARM.

**Figure 6 sensors-22-07820-f006:**
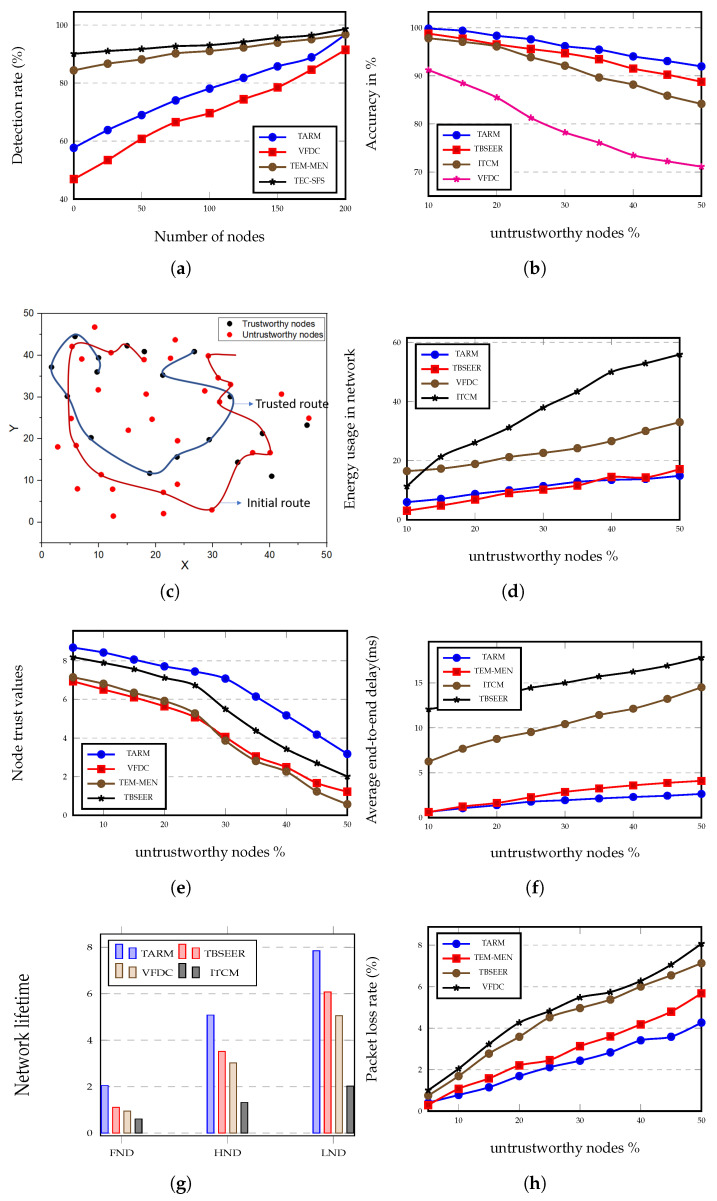
Simulation results. (**a**) Percentage detection rate versus the number of nodes in the network. (**b**) Accuracy in % with respect to the increasing percent of untrustworthy nodes. (**c**) Representation of trusted route and initial route paths for 50 nodes in $50\times 50\;m^{2}$ network size. (**d**) Evaluation of energy consumption (mJ) of all algorithms for varying percentages of untrustworthy nodes. (**e**) Trust values of nodes for different untrustworthy node percentages. (**f**) Average end-to-end delay in ms versus the percent of untrustworthy nodes. (**g**) Network lifetime evaluation for FND, HND, and LND. (**h**) Rate in % for packet loss in the network.

**Figure 7 sensors-22-07820-f007:**
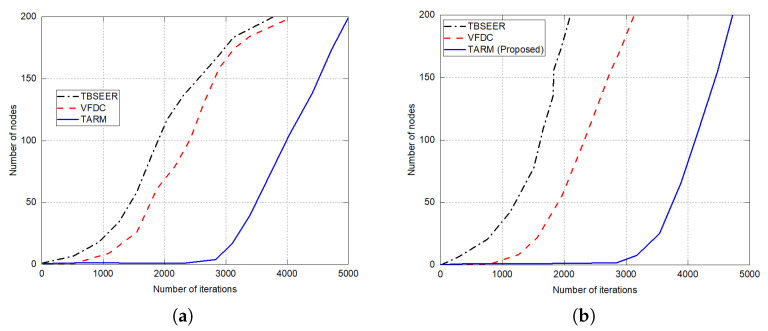
Network lifetime evaluation for untrusted scenarios: (**a**) 25% of untrustworthy nodes out of the total 200 nodes; (**b**) 50% of untrustworthy nodes out of the total 200 nodes.

**Figure 8 sensors-22-07820-f008:**
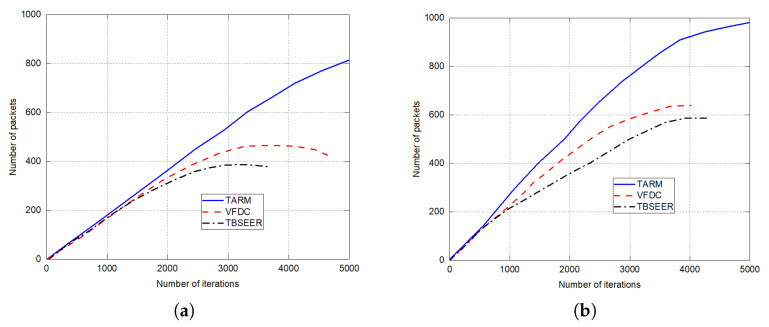
Throughput analysis for untrusted scenarios: (**a**) 25% of untrustworthy nodes out of the total 200 nodes; (**b**) 50% of untrustworthy nodes out of the total 200 nodes.

**Figure 9 sensors-22-07820-f009:**
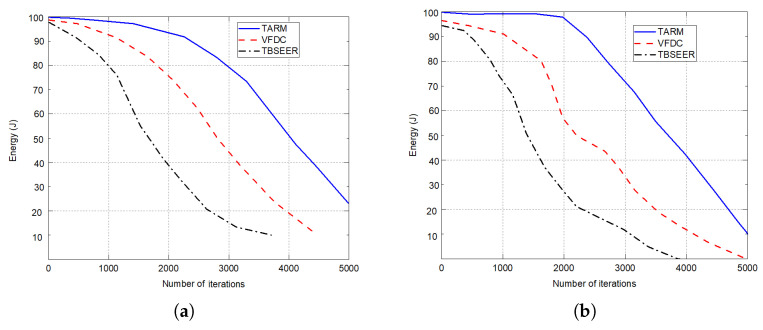
Energy efficiency in joules for untrusted scenarios: (**a**) 25% of untrustworthy nodes out of the total 200 nodes; (**b**) 50% of untrustworthy nodes out of the total 200 nodes.

**Table 1 sensors-22-07820-t001:** Summary of related work on recently published trust-aware protocols.

Reference	Algorithm/Protocol	Advantages	Remarks
Wang et al. [16]	TEM-MEN	Reliable because of efficient hostile attacks prevention	Failed to minimize the energy and costly due to multiple mobile edge nodes
Hu et al. [17]	TBSEER	Resolve the congestion and serious energy dissipation problems	Increasing the percentage of malicious nodes increases the number of rounds needed to exclude them.
Wang et al. [10]	VFDC	Overcomes limited storage capacity and weak computation power due to mobile edge data collector	The applications complying with the edge intelligence network have to be made more trustworthy
Yang et al. [18]	ITCM	Improves detection accuracy of malicious nodes and solves	Not suitable for resource constraint systems, convergence speed needs to be improved
[19,20,21]	Trustworthy nodes selection	Can be improved to encounter other external or internal attacks such as ballot stuffing, conflicting behavior, Sybil attacks, etc.	Security breach from distributed network threats, not cost effective
Zhang et al. [8]	DRL	Intelligent trust nodes selection	High computational complexity and resource rich
Majhi et al. [22]	ALO-K	Efficient hybrid clustering	Intra-cluster distance must be minimum and F-measure must be maximum for better cluster quality
Das et al. [23]	Modified BCO with K-means	Low convergence time	Restricted to single-objective optimization functions
Khan et al. [24]	Modified ABC	Detects Sybil attacks in pervasive edge computing based IIOT	
Kumar et al. [25]	ACO	Efficient data collection by minimizing delay and maximizing the network life	Hard to decide population size and initial parameters
Rm. et al. [26]	PCA		
Iwendi et al. [27]	WOA-SA	Efficient selection of cluster heads	Energy consumption and security issues need to be addressed
**Our approach**	**TARM**	High convergence with no local optima problem	None

**Table 2 sensors-22-07820-t002:** Frequently used notations.

Symbols	Meaning
pop	population of gray wolf
*p*	current iteration
pmax	maximum iterations
*W*	swarm size
*R*	maximum communication range
$itr_{c}$	current iteration
*t*	number of cycles or iterations
$f_{i}$	objective function value
onl	onlooker bees
Emp	employer bees
$F_{ij}$	*i*th dimensional data of *j*th food site
*S*	sink node
$F_{j}^{max}$	upper limit for $F_{ij}$
$F_{j}^{min}$	lower limit for $F_{ij}$
$F_{j}$	*j*th site for food.
${RG}_{CH}$	relay group for packet forwarding, it yields minimum value
$F_{kj}\in \left[1,k\right]$	food source searched randomly
*k*	number of food sites.
φ	random number which lies in [−1, 1]
${DD}_{j}$	rate of data delivered from *j*th energy to ‘s’ queue length
$\left(w_{j}\right)$	weight of routing
${ch}_{j}$	cluster head for *j*th cluster
${ch}_{i}$	cluster head for *i*th cluster
$DD({CH}_{m})$	likelihood of delivery for transmission of data
$DD{\left({CH}_{m}\right)}^{\prime }$	indicates upgraded node model for delivery.

**Table 3 sensors-22-07820-t003:** Network parameters.

Parameters	Value
Deployment area	$50\times 50\;m^{2}$
Total nodes	200
Initial energy	2 J
Packet size	1000
Number of rounds	5000
Threshold distance of transmission	20 m
CHs percent	10–15%

**Table 4 sensors-22-07820-t004:** Average route length.

Algorithms	Average Route Length
VFDC	4.41
TBSEER	4.0
TEM-MEN	3.79
TARM	3.22

**Table 5 sensors-22-07820-t005:** Intra-cluster parameters comparison with other optimization approaches used for clustering of nodes.

Methods	A ^1^	B ^2^	C ^3^
PSO	1124	0.89	2.3
Whale optimization	1256	0.45	1.0
Cuckoo search	1466	0.73	1.7
GWO	1689	0.38	1.5
Modified GWO	1858	0.26	0.48

^1^ Total packets sent to CH. ^2^ Average Energy usage (intra-cluster) in joules. ^3^ Cluster balancing factor.

**Table 6 sensors-22-07820-t006:** Success rates and average time of execution for all algorithms.

Algorithms	Average Time	Success Rate
PSO	60.12	68
Hybrid WGWO	57.44	70
ACO	42.78	59
MBCO	35.05	87
ALO-K	31.84	92
TARM	28.56	97

**Table 7 sensors-22-07820-t007:** Dataset description.

Datasets	Number of Instances	Classes	Number of Features
Iris [36]	150	3	4
Wine [37]	178	3	13
RSSI [38]	6611	3	15
Dry bean [39]	13,611	7	17
Air quality [40]	9358	2	15

**Table 8 sensors-22-07820-t008:** Average ranks of algorithms based on optimized values.

Datasets	PSO	HybridWGWO	ACO	MBCO	ALO-K	TARM
Iris	113.7	96.1	101.4	95.4	97.4	97.25
	(6)	(2)	(5)	(1)	(4)	(3)
Wine	16,960.8	16,480.1	16,960.8	16,230.9	16,334.5	16,210.1
	(5.5)	(4)	(5.5)	(2)	(3)	(1)
RSSI	4763.1	4496.1	4440.8	4440.8	4432.6	4225.7
	(6)	(5)	(3.5)	(3.5)	(2)	(1)
Dry bean	1995.4	1988.9	1967.6	1948.2	1901.6	1948.2
	(6)	(5)	(4)	(2.5)	(1)	(2.5)
Air quality	2989.5	2701.3	2843.6	2248.6	2248.6	2081.6
	(6)	(4)	(5)	(2.5)	(2.5)	(1)
(Average ranks) $r_{i}$	**5.9**	**4.0**	**4.6**	**2.3**	**2.5**	**1.7**

**Table 9 sensors-22-07820-t009:** Average ranks of algorithms based on optimized values.

Algorithm	*z*-Value	*p*-Value	α/(y−i)	Hypothesis
PSO	−3.54969	0.00020	0.01	Rejected
Hybrid WGWO	−1.94388	0.02619	0.0125	Accepted
ACO	−2.45098	0.00714	0.0166	Rejected
MBCO	−0.50709	0.30854	0.025	Accepted
ALO-K	−0.67613	0.25143	0.05	Accepted

## Data Availability

The data used for this paper are cited within the article.

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
