# Peer review of "Trust-Aware Routing Mechanism through an Edge Node for IoT-Enabled Sensor Networks"

_sensors, 2022, doi:10.3390/s22207820_

Round 1
Reviewer 1 Report
1. The motivation of why a trust-aware routing mechanism is needed is not clear in abstract which should be addressed.
2. The related work just presented the peers' work without discussion. It will be better if they can point out the advantages and the disadvantages of the peers' work.
3. There is no reference about the real dataset. Authors are requested to provide the references for the real dataset
4. Have you ever considered that communication errors may cause fluctuations of connections which may cause the trusted nodes to lose trustworthiness?
5. In the introduction, the first contribution listed in this paper is unclear. What effects does the modified grey wolf optimization approach have?
Similarly, for the listed fourth contribution, after the performance analysis, where is the superiority of this solution in each index?
6. The conclusion can be made more technically strong.
Author Response
Technical comments:
- The motivation of why a trust-aware routing mechanism is needed is not clear in abstract which should be addressed.
Response to Comment 1.
Thank you very much for your comment. To honour the suggestion, the abstract is modified and contains dew lines that clearly mentions the motivation behind the work. The lines are added at the beginning of the abstract and are highlighted in blue colour in the manuscript.
- The related work just presented the peers' work without discussion. It will be better if they can point out the advantages and the disadvantages of the peers' work.
Response to Comment 2.
Thanks for your comment. To honour your suggestion, a table is introduced in the related works section that presents recent relevant works along with their technical analysis, advantages and disadvantages.
- There is no reference about the real dataset. Authors are requested to provide the references for the real dataset
Response to Comment 3.
Thanks for your precise comment. We are very sorry for this mistake. It is corrected in the revised paper. The relevant references are cited in the results section.
- Have you ever considered that communication errors may cause fluctuations of connections which may cause the trusted nodes to lose trustworthiness?
Response to Comment 4.
Thank you very much for your comment. The communication errors here in our proposed work are in terms of Packet drop rate and Average end to end delay. However, in some scenarios where the edge computing is completely implemented, the issues like computation cost and communication overhead are treated in a serious manner. As this paper deals only with one edge node responsible to gather trust data, the communication errors relating to trustworthiness of nodes are negligible and can be removed as explained in the results section. However, the future aspect to this model that implements the edge computing technique must consider the communication errors such as communication overhead and cost of computation.
- In the introduction, the first contribution listed in this paper is unclear. What effects does the modified grey wolf optimization approach have?
Response to Comment 5.
Thanks for your comment. We want to justify this by stating that in the original GWO, half of the iterations are devoted to exploration and the other half are dedicated to exploitation, which affects the accuracy in the calculation of global optima. However, modified GWO is motivated by social hierarchy and hunting behaviour of grey wolves. It also uses a fitness function effectively that avoids the local optima values.
- For the listed fourth contribution, after the performance analysis, where is the superiority of this solution in each index?
Response to Comment 6.
Thanks for your comment. To honour the suggestion, kindly have a look at figures 6(a), 6(b), 6(d), 6 (f), 6(g) in the results section. These figures give the comparison analysis among some recent similar methods and our method. These figures show that our method is superior to other methods in terms of accuracy, detection rates, energy consumption, delay and network life.
- The conclusion can be made more technically strong.
Response to Comment 7.
Thanks for your comment. To honour the suggestion, we have modified the conclusion section. The modifications are highlig
Reviewer 2 Report
The authors of this study proposed "rust aware routing mechanism through an edge node for IoT-enabled sensor networks" The topic is interesting and well-formulated. It can be considered for publication with the following minor modifications:
-The introduction section needs improvement by including more information to justify the proposed work.
-The literature review is carried out poorly. The authors must be critical by stressing the advantages and disadvantages of the existing schemes in order to differentiate the proposed study.
-In terms of resolutions, Figure 2 is nearly undetectable. The authors need to redraw the diagram.
Author Response
- The introduction section needs improvement by including more information to justify the proposed work.
Response to Comment 1.
Thank you very much for your comment. To honour the suggestion, we have modified the introduction section and have included justification why our proposed work is better than other similar works.
- The literature review is carried out poorly. The authors must be critical by stressing the advantages and disadvantages of the existing schemes in order to differentiate the proposed study.
Response to Comment 2.
Thanks for your precise comment. To honour the suggestion, the literature review is revised and some recent relevant works are added in it. The technical analysis, advantages and disadvantages of the works are discussed in a tabular form that is highlighted in the manuscript.
- In terms of resolutions, Figure 2 is nearly undetectable. The authors need to redraw the diagram.
Response to Comment 3.
Thanks for your comment. To honour the suggestion, Figure 2 is redrawn and is included in the manuscript with good visibility and resolution.